# Multi-Response Optimization of Porous Asphalt Mixtures Reinforced with Aramid and Polyolefin Fibers Employing the CRITIC-TOPSIS Based on Taguchi Methodology

**DOI:** 10.3390/ma12223789

**Published:** 2019-11-18

**Authors:** Carlos J. Slebi-Acevedo, Pablo Pascual-Muñoz, Pedro Lastra-González, Daniel Castro-Fresno

**Affiliations:** GITECO Research Group, Universidad de Cantabria, Avda. de Los Castros s/n., 39005 Santander, Spain; carlosjose.slebi@unican.es (C.J.S.-A.); pablo.pascualm@unican.es (P.P.-M.); pedro.lastragonzalez@unican.es (P.L.-G.)

**Keywords:** porous asphalt, fibers, taguchi, critic, topsis

## Abstract

For the optimum design of a Porous Asphalt (PA) mixture, different requirements in terms of functionality and durability have to be fulfilled. In this research, the influence of different control factors such as binder type, fiber content, and binder content were statistically investigated in terms of multiple responses such as total air voids, interconnected air voids, particle loss in dry conditions, particle loss in wet conditions, and binder drainage. The experiments were conducted based on a Taguchi L18 orthogonal array. The best parametric combination per each response was analyzed through signal to noise ratio values. Multiple regression models were employed to predict the responses of the experiments. As more than one response is obtained, a multi-objective optimization was performed by employing Criteria Importance through Criteria Inter-Correlation (CRITIC) and Technique for Order Preference by Similarity to Ideal Solution (TOPSIS) methodologies. The weights for the selection of the functional and mechanical performance criteria were derived from the CRITIC approach, whereas the ranking of the different experiments was obtained through the TOPSIS technique. According to the CRITIC-TOPSIS based Taguchi methodology, the optimal multiple-response was obtained for a polymer modified binder (PMB) with fiber and binder contents of 0.15% and 5.0%, respectively. In addition, good results were obtained when using a conventional 50/70 penetration grade binder with a 5.0% binder content and 0.05% fiber content.

## 1. Introduction

In the last 20 years, the use of porous asphalt (PA) mixtures in wearing courses has increased considerably around the world due to the multiple advantages that this type of hot mix asphalt (HMA) offers [1]. This mixture is characterized by the predominant use of high quality open-graded crushed coarse aggregates along with a small amount of fine aggregates in order to obtain a stone-on-stone contact and high interconnected air voids [2]. As a result, the granular skeleton formed is capable of resisting permanent deformation, whereas the connected voids allow the water to be evacuated from the surface of the pavement. Besides, when the water is removed, the splash and spray is minimized, as well as the aquaplaning effect [3]. Other advantages include the improvement of the pavement friction, especially in wet conditions, mist attenuation on rainy days, mitigation of the urban heat island effect, and enhancement of the surface reflectivity, especially in nighttime [4,5].

The porous structure of the PA mixture also contributes to mitigating the noise generated by the traffic loads [6]. In fact, porous asphalt pavements are currently the most widely used pavements worldwide when it comes to the reduction of the traffic noise [6,7,8,9]. As suggested by other researchers [6,10,11], the connected porous structure helps to dissipate the sound energy, whereas the surface pores and the macrotexture contribute to limit noise generation phenomena (i.e., air pumping or air sucking) in the tire-road contact. 

Despite their multiple benefits, the high voids content makes the open graded mixtures prone to suffer raveling [5], which can be defined as the loss of aggregate on the top of the surface during the service life of pavement structure [12]. Moreover, due to their high porosity, a lower mortar content is present in PA mixtures when compared to dense graded mixtures and hence, the adhesion between binder and aggregates is worse. Similarly, as the mixture is highly exposed to the air and the wet conditions of the environment, the binder film is susceptible to oxidation and consequently, the strength of the binder-aggregate bonding is affected severely. 

In order to improve the durability of the mix, several agencies around the world have employed different admixtures. Open graded friction course (OGFC) mixtures, as they are called in the United States (US), began to be used in the 1970s in response to a Federal Highway Administration program (FHWA) to increase the frictional resistance on surface courses [13]. However, the applicability of OGFC mixtures was relatively low until the 1980s, when the mix designs were improved by using polymer modified binder (PMB) and fiber additives to stabilize the mix and prevent the drain down [4]. Similarly, China began to apply porous asphalt courses in the 1980s. Nowadays, high-viscosity modified forms of asphalt binder are used [5] for that purpose. Regarding Europe, Spain was one of the first countries that focused on the study of PA mixtures [1,14]. In the 1980s, the University of Cantabria carried out a study based on developing a design and control methodology [14]. As a result, the Cantabro test to evaluate the particle loss [15,16] was developed and started to form part of the European standard methods (EN 12697-17). Also during that period, the employment of porous asphalt mixtures as wearing course in The Netherlands became very popular and widely used not only due to the road safety aspects, but also because of the potential to mitigate the noise pollution from the traffic loads [17]. In this country, the modified binders are only employed for special purposes [1]. Although the general tendency in Europe is towards the use of modified binders as they possess higher flexibility and lead to thicker binder films with no binder drainage [18], other researchers suggest that there is a lack of information proving the higher durability of the PA mixtures using PMB [19]. In addition, although PMB brings ductility to the mixture due to the elastic recovery properties and let the binder content to be increased [20], the use of additives such as fibers has attracted much attention as it could prevent the draining of the binder while improving the mix durability [21,22,23]. 

Several types of fibers have been used in hot asphalt mixtures: cellulose, polyester, carbon, basalt, glass, polyacrylonitrile, nylon, or aramid, among others [24,25,26,27,28,29,30]. Asphalt concrete (AC) is the type of mixture where the use of fibers as a reinforcement has been extensively used [26]. For example, Tapkin et al. [31] reported 20% higher Marshall stability values when adding 0.3% polypropylene fibers by weight of aggregates. Xu et al. [32] reported that polymer fibers such as polyester and polyacrylonitrile have greater effects on the resistance to permanent deformation, fatigue life, and indirect tensile strength in comparison to lignin and asbestos fibers. Similarly, the authors suggested an optimum fiber content of 0.35% by mass of mixture in order to achieve the best performance outputs with respect to rutting resistance and indirect tensile strength. Takaikaew et al. [33] performed a detailed laboratory experimental plan including Marshall stability, indirect tensile strength and stiffness modulus, resilient modulus, dynamic creep, indirect tensile fatigue, and rutting resistance tests on asphalt concrete mixtures with different types of binder (conventional, rubber modified asphalt and polymer modified asphalt) and polyolefin/aramid fibers. According to the results, the addition of 0.05% of fibers by weight of mixtures considerably improved the mechanical performance of the mixture, regardless of the asphalt binder type used. Similarly, Kaloush et al. [34] reported that polypropylene/aramid fibers notably enhanced the mixture’s performance against rutting resistance, fatigue, and thermal cracking. Regarding the PA mixtures, cellulose fibers have become the most common stabilizer additive [21,35,36,37]. Lopes et al. [21] evaluated the performance of porous asphalt mixtures having cellulose fibers and polymer modified binder. The authors concluded that cellulose fibers enables the increase of the binder content by providing proper retention, thus resulting in greater aggregates coating and improved durability of the mix. Similar results were obtained by Valeri et al. [36], who assessed the durability of a PA mixture incorporating cellulose fibers but using a conventional 50/70 penetration grade bitumen instead of a modified binder. 

While good mechanical performance has been observed when using polyolefin/aramid (POA) fibers in asphalt concrete mixes, the use of this fiber type has not been tested in PA mixtures. Additionally, many studies have focused on the effects of fibers in only one category of bitumen, either a conventional binder or a polymer modified binder, but not both. Likewise, the use of fibers has only been valued as a stabilizer additive and not as a reinforcement additive. Besides, the design of a porous asphalt mixture reinforced with fibers requires optimum binder and fiber contents that guarantees an adequate resistance to raveling and to the harmful action of the water, the absence of binder drainage, and a big enough air voids content to enable the water to be removed from the surface and reduce the rolling noise. 

In order to comply with the aforementioned, POA fibers are here presented as an alternative additive for the stabilization of the mixtures and the improvement of their raveling resistance with no harm of their optimal functionality. Furthermore, the novel CRITIC-TOPSIS based on Taguchi optimization technique is proposed for the design of porous asphalt mixtures with the aim of finding out the most relevant input parameters from the standpoint of their functionality and durability. In other words, the relationship between type of binder, fiber content, and binder content are considered as the main control factors to estimate the optimal solution for the mixture. As dependent variables or responses, total air voids, interconnected air voids, raveling resistance in dry conditions, raveling resistance in wet conditions, and binder drain down are considered.

The paper begins with an introduction section where the literature review of previous related research works, scope and objectives of this study are referred. This section is followed by a detailed explanation of the CRITIC-TOPSIS employed here based on the Taguchi novel technique. Materials and research methods are thoroughly described in the third section, including material properties, sample preparation and experimental testing plan. Results and discussion in section four describes main findings and includes the statistical analysis performed and the different regression models aimed at predicting the response values. The transformation of the multi-response into a single response through the CRITIC-TOPSIS approach is also described. Finally, the main conclusions are drawn in the last section. 

## 2. Experimental Design

### 2.1. Taguchi Method

The Taguchi method has been considered by other researchers as an efficient statistical method to optimize the analysis of experimental variables and improve the accuracy of the responses [38,39]. Additionally, this method estimates the contribution of individual control factors that influence the quality of a design process or optimum mix [40]. Although initially developed to improve the quality of manufactured products, its use was extended to the civil engineering field [41,42,43,44,45].

In this study, the design of experiments was carried out according to the Taguchi L_18_ full factorial orthogonal array (2^1^ × 3^2^) in order to investigate the relationship between different binder and fiber contents for different types of binders. Their effects on the durability and functionality of the PA mixture were also analyzed. 

The signal to noise ratio (SNR) is a measure that enables the determination of significant input parameters by assessing the minimum variance [42]. In other words, higher values of SNR suggest more relevance of the input parameters on the responses. In general SNR can be specified in three different scenarios namely the *smaller-the-better*, the *larger-the-better*, and the *nominal-the-better*. In this research, the smaller-the-better scenario is employed to minimize the loss of particles in dry and wet conditions as well as the binder drainage, while the larger-the-better is employed to maximize the total air and interconnected air voids. The equations used for calculating the *smaller-the-better* and the *larger-the-better* scenarios are Equations (1) and (2), respectively: (1)$$\frac{S}{N}=\;-10\log_{10}\left(\frac{1}{n}\overset{n}{\underset{i=1\;}{\sum }}y_{i}^{2}\right)$$
(2)$$\frac{S}{N}=\;-10\log_{10}\left(\frac{1}{n}\overset{n}{\underset{i=1\;}{\sum }}\frac{1}{y_{i}^{2}}\right)$$
where $y_{i}$ corresponds to the experimental result at the *i*th experiment and *n* refers to the total number of experiments [42]. Binder type (50/70, PMB), Fiber content (*FC*) and binder content (*BC*) were selected as control input parameters and their corresponding levels were determined as shown in Table 1. Thus, 18 sets of experiments with three replicates per design were carried out. Table 2 presents the L18 mixed orthogonal array for conducting the design of experiments.

### 2.2. Technique for Order of Preference by Similarity to Ideal Solution (TOPSIS)

The TOPSIS approach is considered one of the most popular mathematical models to determine the optimal solution of a multi-criteria decision-making analysis (MCDM). In civil engineering, TOPSIS is considered the second most popular multi-criteria technique right after the analytic hierarchy process (AHP) [46]. Zhang et al. [47] evaluated public transport priority performance by applying TOPSIS. Jato et al. [48] implemented a hybrid decision support model incorporating TOPSIS to rank different wearing courses in highly trafficked European roads. On another study, Egle and Jurgita [49] ranked many alternatives in order to improve the daylighting in vernacular buildings.

Unlike in previous investigations, in this research TOPSIS was adopted to transform the multi response problem resulting from the design of experiments into a single response problem, thus giving the best set of alternatives. Total air voids, interconnected air voids, particle loss in dry conditions, particle loss under the influence of water, and binder drainage were considered to be the quality criteria required for TOPSIS to set those reinforced porous asphalt alternatives. 

The algorithm of TOPSIS is structured on the basis of the concept of distance of the alternatives proposed to positive and negative ideal solutions [50]. In other words, a positive ideal solution (PIS) refers to an alternative that maximizes the benefit responses and minimizes the cost responses, whereas a negative ideal solution (NIS) is considered the least preferred solution as it minimizes the benefit responses and maximizes the cost responses. Therefore, the best alternative would be the one closest to the positive ideal solution and furthest from the negative ideal solution [51]. 

Following, the steps involved in the TOPSIS technique are presented. 

**Step 1.** Build the decision-making matrix, with alternatives representing input parameters from the manufacturing of asphalt mixes and criteria (or attributes) corresponding to the responses generated by the experimental results. In line with this, the matrix can be expressed as follows: (3)$$D=\;\left(\begin{array}{cccccc} p_{11} & p_{12} & \ldots & p_{1j} & \ldots & p_{1n}\\ p_{21} & p_{22} & \ldots & p_{2j} & \ldots & p_{2n}\\ \vdots & \vdots & \ddots & \ldots & \ldots & \vdots \\ p_{i1} & p_{i2} & \vdots & p_{ij} & \ldots & \vdots \\ \vdots & \vdots & \vdots & \vdots & \ddots & \vdots \\ p_{m1} & p_{m2} & \cdots & p_{mj} & \cdots & p_{mn} \end{array}\right)$$
where $p_{ij}$ corresponds to the performance of the *i*th experimental alternative with respect to the *j*th attribute. 

**Step 2.** Normalize the decision matrix as follows:(4)$$r_{ij}=\;\frac{p_{ij}}{\sqrt{\sum_{i=1}^{m}p_{ij}^{2}}},\;\;\;i=1,2,3,\ldots ,m\;\;\;j=1,2,3,\ldots ,n$$
where $r_{ij}$ refers to the normalized rating of the attribute. In this step, various attribute dimensions are transformed into non-dimensional attributes in order to make possible the comparisons across the responses. 

**Step 3.** Calculate the weighted normalized decision matrix as follows:(5)$$\left[v_{ij}\right]=\left[w_{j}r_{ij}\right]$$
where $\left[v_{ij}\right]$ corresponds to the weighted normalized matrix and $w_{j}$ refers to the weightage of the *j*th criterion. The following should be fulfilled:(6)$$\sum_{j=1\;}^{n}w_{j}=1.$$

**Step 4.** Calculate the positive (PIS) and negative ideal solutions (NIS). 

The positive ideal solution is determined as follows:(7)$$V{\;}^{+}=\left(v_{1}^{+},v_{2}^{+},v_{3}^{+},\ldots v_{n}^{+}\right)=\left\{\left(max\;v_{ij}|j\in I\right),\;\left(min\;v_{ij}|j\in J\right)\right\}$$

The negative ideal solution is determined as follows:(8)$$V^{-}=\left(v_{1}^{-},v_{2}^{-},v_{3}^{-},\ldots v_{n}^{-}\right)=\left\{\left(min\;v_{ij}|j\in I\right),\;\left(max\;v_{ij}|j\in J\right)\right\}$$
where I is related with beneficial criteria and J with non-beneficial criteria; i=1,2,…,m; and j=1,2,…,n.

**Step 5.** Determine the distance of each alternative from positive and negative ideal solutions.

The distance to the positive ideal solution is as follows:(9)$$S_{i}^{+}=\;\sqrt{\sum_{j=1}^{n}{\left(v_{ij}-v_{j}^{+}\right)}^{2},\;}i=1,2\ldots ,m.$$

The distance to the negative ideal solution is as follows:(10)$$S_{i}^{-}=\;\sqrt{\sum_{j=1}^{n}{\left(v_{ij}-v_{j}^{-}\right)}^{2},\;}i=1,2\ldots ,m.$$

**Step 6.** Calculate the relative closeness from each alternative to the positive ideal solution:(11)$$C_{c}^{*}=\;\frac{d_{i}^{-}}{d_{i}^{-}+d_{i}^{+}}$$
where $C_{c}^{*}$ is the relative closeness coefficient; $i=1,2,\ldots ,m;\;0\leq C_{c}^{*}\leq 1$. 

**Step 7.** Rank the different alternatives and select the option with $C_{c}^{*}$ closest to 1. 

### 2.3. Criteria Importance through Inter-Criteria Correlation (CRITIC)

When multiple responses are involved in a decision-making problem, prioritizing one criterion against the others turns out to be a complex task due to the nature of subjectivity. To avoid that, the CRITIC methodology developed by Diakoulaki et al. [52] arose as an innovative approach in the category of Multi-Objective Decision Making (MODM) methods. Based on this methodology, weights of relative importance can be determined in an objective manner as correlated to certain criteria [53]. This has been applied in different areas of the engineering as a decision support system, including manufacturing processes, supply chain, and risk management [54,55]. As for the combination of design of experiments and multi-criteria decision-making analysis, no research has been carried out so far, with responses being commonly assigned based on criteria with equal weightage [56]. Therefore, this research seeks to employ a novel approach by means of using a technique that does not require human participation and helps to automatize decision making, along with the TOPSIS method, which enable going from a multi-response problem to an optimized single response. Following this, a brief description of the CRITIC technique is presented based on reference [52]. 

**Step 1.** Define the finite set *A* of *n* alternatives with respect to *m* evaluation criteria as follows: (12)$$A={\left[a_{ij}\right]}_{n*m}=\;\left[\begin{array}{cccc} a_{11} & a_{12} & \cdots & a_{1m}\\ a_{21} & a_{22} & \cdots & a_{2m}\\ \vdots & \vdots & \ddots & \vdots \\ a_{n1} & a_{n2} & \cdots & a_{nm} \end{array}\right]\;(i\;=\;1,2,\ldots ,n\;\mathrm{and}\;j\;=\;1,2,\ldots m)$$
where $a_{ij}$ represents the response value of the *i*th alternative on the *j*th criterion.

**Step 2.** Normalize the decision matrix using the following equation:(13)$${\bar{a}}_{ij}=\;\frac{a_{ij}-a_{j}^{worst}}{a_{j}^{best}-a_{j}^{worst}}$$
where ${\bar{a}}_{ij}$ is the normalized performance value of the *i*th alternative for the *j*th criterion, $a_{j}^{best}$ corresponds to the best performance value for *j*th criterion, and $a_{j}^{worst}$ is the worst performance value for *j*th criterion. 

**Step 3.** Calculate the standard deviation σ  of each vector $a_{j}$, which quantifies the contrast intensity of the corresponding criterion. 

**Step 4.** Build the symmetric *m × m* matrix with the generic element $r_{jk}$, which corresponds to the linear correlation coefficient between vectors $a_{j}$ and $a_{k}$.

**Step 5.** Determine with the following formula the measure of the conflict created by criterion *j* with respect to the decision situation defined by the rest of the criteria:(14)$$\sum_{k=1}^{m}1-r_{jk}.$$

**Step 6.** Calculate $C_{j}$, which represents the quantity of information contained in *j*th criterion: (15)$$C_{j}=\sigma \;*\;\overset{m}{\underset{k=1}{\sum }}1-r_{jk}$$

**Step 7.** Determine the objective weights of the *j*th criterion:(16)$$W_{j}=\;\frac{C_{j}}{\sum_{k=1}^{m}C_{j}}$$

## 3. Materials and Methods

### 3.1. Materials

In this study, ophite and limestone were used as coarse and fine aggregates, respectively, for the manufacturing of the PA mixtures. Limestone was also employed as filler material. The gradation curve corresponds to a PA mixture with nominal maximum aggregate of 16 mm commonly known as PA16 by Spanish specifications [57]. The physical properties and gradation of aggregates can be seen in Table 3 and Figure 1, respectively. As for the bituminous binder, in this research a conventional 50/70 penetration grade bitumen (50/70) and a polymer modified binder (PMB 45/80-65) were used. The main properties of the binders are shown in Table 4.

Regarding the fibers, a blend of polyolefin and aramid synthetic fibers (POA) was used for both improving the durability of the PA mixture and as a stabilizing additive. The density of the blend according to the standard method UNE-EN 1097-6 is 0.947 g/cm^3^. The main physical properties of the POA fibers and a picture of them can be seen in Table 5 and Figure 2, respectively. 

### 3.2. Manufacturing of the Porous Asphalt Sample

For the manufacturing of the PA samples using conventional 50/70 penetration grade bitumen, coarse and fine aggregates and the filler were first heated for six hours in an oven at 170 °C and then thoroughly mixed with the fibers. Afterwards, the binder at 150 °C was placed into the mixture and continuously blended until the combination fiber-aggregate was well coated. When the polymer modified binder was used, the aggregates and binder temperatures increased from 170 °C to 185 °C and from 150 °C to 165 °C, respectively. Finally, all the test samples were compacted by 50 blows per side according to the EN 12697-30. 

### 3.3. Laboratory Testing Plan

In order to optimize the functionality and durability of the PA mixture, total air voids, interconnected air voids, binder drainage, and raveling resistance in dry and wet conditions have been considered as porous asphalt quality criteria. Based on the volumetric properties test [58,59], total air voids ($T_{AV}$) and interconnected air voids ($I_{AV}$) were calculated following the Equations (17) and (18), respectively:(17)$$T_{AV}\left(\%\right)=\left(1-\;\frac{m}{V*G_{mm}}\right)*100\%$$
(18)$$I_{AV}\left(\%\right)=\;\frac{V-\frac{m-m_{w}}{\rho_{w}}}{V}*100\%$$
where *m* corresponds to the mass of the specimen in the air; *V* refers to the total volume of the specimen, which is calculated geometrically; $G_{mm}$ is the theoretical maximum specific gravity of the mixture; and $m_{w}$ is the saturated specimen mass in water. 

To assess the durability of the PA mixture in terms of its raveling resistance, the Cantabro loss particle test (EN 12697-17) was carried out. According to this test, the particle loss refers to loss mass of a PA specimen after applying 300 revolutions in the Los Angeles abrasion machine. The particle loss (*PL*) is calculated as follows:(19)$$PL\;\left(\%\right)=\;\frac{w_{1}-w_{2}}{w_{2}}*100\%\;\;$$
where $w_{1}$ is the initial weight of the specimen and $w_{2}$ refers to the final weight of the specimen. 

Additionally, the Cantabro test in wet conditions was performed following the Spanish standard method NLT 362/92. Before the test, specimens were conditioned by submerging them in water at 60 °C for 24 h and then exposed to air at 25 °C for another 24 h.

To assess the stability of the mixture, the mesh basket binder drain down test according to the EN 12697-18 standard was used. The test consists of quantifying the material lost by drainage after 3h at the test temperature [60]. The binder drainage (*BD*) in percentage is determined as follows: (20)$$BD\;\left(\%\right)=\;\frac{m_{2}-m_{1}}{1100+B}*100$$
where $m_{1}$ is the initial mass of the tray and foil, $m_{2}$ refers to the mass of the tray and foil including the drained material, and B corresponds to the initial mass of the binder in the mixture. 

The experimental part was developed in the roads laboratory of the University of Cantabria. The structured framework of the multi-objective optimization can be observed in Figure 3. 

## 4. Results and Discussion

### 4.1. Analysis of Signal to Noise Ratios (SNR) and Means on Different Responses

The different responses obtained by way of the Taguchi L18 orthogonal array can be observed in Table 6. Total and interconnected air voids are considered important parameters to assess the functionality of the PA mixture in terms of permeability, noise properties and macrotexture [61]. As for the results, mean values of $T_{AV}\;$ and $I_{AV}$ ranged from 17.50% to 23.20% and 11.20% to 17.26%, respectively (Table 6). Similarly, a direct relation exists between both responses, with a Pearson correlation coefficient of 89%. Following the Taguchi methodology, $T_{AV}$ and $I_{AV}$ were converted into signal-to noise ratio (SNR). The highest values of total and interconnected air voids are very important for improving the functional performance of the mixture. Therefore, the *larger-the-better* equation was employed for calculating the SNR. Figure 4 and Figure 5 show the main effect of the SNR and the means for the total and interconnected air voids, respectively.

A SNR analysis of the effect of the input factors, i.e., binder type (BT), fiber content (*FC*) and binder content (*BC*), on the total and interconnected air voids was carried out (Figure 4 and Figure 5). SNR makes it possible to show the optimal levels of the different input factors for the optimal responses ($T_{AV}$ and $I_{AV}$). As an example, the levels and SNR for the factors giving the best $T_{AV}$ response are: level 2 and SNR = 26.03 for BT factor; level 1 and SNR = 26.07 for *FC* factor; and level 1 and SNR = 26.49 for *BC* factor. Therefore, the optimum $T_{AV}$ can be obtained by using a polymer modified binder, with the lowest binder content and no fibers. Despite that, it is worth mentioning that the binder content is the input factor that most influences the change in the air voids value in comparison to the binder type or fiber content, as can be observed in Figure 4b and Figure 5b. On the other hand, the type of binder does not have a notable influence on the $T_{AV}$ response. 

Concerning the evaluation of the mechanical performance, raveling resistance was evaluated on Marshall Samples in dry and wet conditions. 

Mean values of the three replicas per design and test along with their corresponding standard deviations can be observed in Table 6. It is also interesting to notice that a direct correlation between the loss particles in dry and wet conditions exists, with a Pearson correlation coefficient of 79%. It means that the lower the values of particle loss in dry conditions ($PL_{DRY})$ are, the lower the values of particle loss in wet conditions $\left(PL_{WET}\right)$ are, too. Figure 6 and Figure 7 depicts the main effects of SNR as well as the means for the loss of particles in dry and wet conditions, respectively. Contrary to the calculation of air voids, the *smaller-the-better* quality characteristics were used to calculate the SNR. The highest value of SNR determines the best level for each control factor. For example, the levels and SNR for the input factors giving the optimal value of $PL_{DRY}$ are: level 2 and SNR = −14.72 for BT factor; level 2 and SNR = −15.90 for *FC* factor; and level 3 and SNR = −13.70 for *BC* factor. This means that the optimum value of $PL_{DRY}$ is obtained when polymer modified binder is used along with 0.05% POA fibers and 5.5% binder content. As for the $PL_{WET}$ value, the highest impact according to SNR values comes from the binder type and the binder content. In fact, the contribution of fibers in terms of raveling resistance under the water action is less appreciable when a polymer modified binder is used, as can be observed in the main effect plots for the means (Figure 7b). 

The non-compacted PA mixtures corresponding to all the designs were subjected to evaluation of their drain down characteristics through the mesh basket drain down test as per the EN 12697-18 standard. Binder drainage (*BD*) results are shown in Table 6. As well as to evaluate the raveling resistance, *smaller-the-better* equation was chosen to calculate the SNR values, as can be seen in Figure 8. According to the results, the levels and SNR values for the factors giving the less binder drainage were: level 1 and SNR = 22.87 for the BT factor; level 3 and SNR = 26.24 for the *FC* factor; and level 1 and SNR = 35.49 for the *BC* factor. In other words, the lowest binder drainage can be obtained when a conventional 50/70 penetration grade binder is used along with 0.15% POA fibers and 4.5% binder content. The reduced value of *BC* (Figure 8b) might suggest that fibers can absorb the free binder in the mix. 

### 4.2. Statistical Analysis of Response Results

The changes in the different responses obtained as a result of the experimental research are shown in Figure 9. The interaction effect between binder content and fiber content is plotted as depending of the binder type per each response value ($T_{AV},\;I_{AV},\;PL_{DRY},\;PL_{WET},\;BD).$ For practical reasons, which are based on the response variable data obtained from tests with mixtures with 50/70 penetration grade binder, an analysis of variance was performed. A 5% significance level and a 95% confidence level were considered for the calculation of the factors affecting the different output parameters (Table 7). The significance of the input parameters in the analysis of variance was identified by comparing the *F*-values of each input parameter. 

Regarding the total and interconnected air voids, binder content (*BC*) has the highest influence, with contribution factors of 82% and 80%, respectively. It means that the binder content in the mixture influences notably its porosity, reducing functional performance characteristics such as permeability and noise generation. On the other hand, fiber content (*FC*) seems not to have a significant effect, probably because the amount of fiber used in this research is too low. Other types of fibers such as the cellulose are able to reduce the amount of voids in the mixture when its content is around 0.3% by weight of mixture, as suggested by other research [36]. However, the *FC* factor does have a higher influence when it comes to the resulting raveling resistance responses, with contributions of 25% and 13% to the particle loss in dry and wet conditions, respectively. As reported by other researchers, fibers in hot mix asphalt act as a reinforcement, forming a three dimensional network inside the mixture [25,26]. In addition, fibers are normally used as stabilizer agents in PA mixtures with high binder contents. The contribution of the fiber content (*FC*) with regard to the binder drainage response is actually approximately 27%. 

In this research, regression analyses were employed for modeling and predicting the response variables. Different models were initially proposed such as linear, linear plus interactions, linear plus squares and full quadratic in order to predict the best response variable. The best fitting models, those with the highest $R^{2}$ values, were finally selected.

The predictive equations obtained from the analysis of the mixtures with 50/70 binder, are given below:(21)$$T_{AV}\left(\%\right)=30.65+114*FC\;\left(\%\right)-2.194\;*BC\;\left(\%\right)\;-21.85\;*FC\;\left(\%\right)*BC\;\left(\%\right)$$
(22)$$I_{AV}\left(\%\right)=25.09+125*FC\;\left(\%\right)\;-2.379*BC\;\left(\%\right)-23.52*FC\left(\%\right)*BC\left(\%\right)$$
(23)$$PL_{DRY}\left(\%\right)=167+169*FC\;\left(\%\right)-57.7*BC\left(\%\right)+635*FC^{2}\left(\%\right)+5.23*BC^{2}\left(\%\right)-47.9*FC\left(\%\right)*BC\left(\%\right)$$
(24)$$Pl_{WET\;}\left(\%\right)=352+649*FC\left(\%\right)\;-119*BC\left(\%\right)-1012*FC^{2}\left(\%\right)+10.13*BC^{2}\left(\%\right)-88.3*FC\left(\%\right)*BC\left(\%\right)$$
(25)$$BD\;\left(\%\right)=27.5+45.6\;*FC\left(\%\right)-12.6\;*BC\;\left(\%\right)+80.4*FC^{2}\left(\%\right)+1.44*BC^{2}\left(\%\right)-12.63*FC\left(\%\right)*BC\left(\%\right)$$

Similarly, the predictive equations obtained from the analysis of the mixtures with PMB 45/80-64 are as follows:(26)$$T_{AV}\left(\%\right)=-11.5\;-72.1\;*FC\;\left(\%\right)+14.3\;BC\;\left(\%\right)+158*FC^{2}\left(\%\right)-1.57*BC^{2}\left(\%\right)+8.7\;*FC\left(\%\right)*BC\left(\%\right)\;$$
(27)$$I_{AV}\left(\%\right)=14.4-28.3*FC\;\left(\%\right)+0.4*BC\left(\%\right)+77*FC^{2}\left(\%\right)\;-0.07*BC^{2}\;\left(\%\right)+1.9*FC\;\left(\%\right)*BC\left(\%\right)$$
(28)$$PL_{DRY}\left(\%\right)=30.51\;-7.47\;*FC\left(\%\right)-4.81*BC\left(\%\right)$$
(29)$$PL_{WET}\;\left(\%\right)=172+32*FC\left(\%\right)-64.4*BC\left(\%\right)+54*FC^{2}\left(\%\right)+6.28*BC^{2}\left(\%\right)-12.0*FC\left(\%\right)*BC\left(\%\right)$$
(30)$$BD\;\left(\%\right)=6.7+12.7\;*FC\;\left(\%\right)-3.28\;*BC\left(\%\right)+51.6\;*FC^{2}\left(\%\right)+0.400*BC^{2}\left(\%\right)-4.50\;*FC\left(\%\right)*BC\left(\%\right)$$

All the regression models for the mixtures using the conventional bitumen fitted very well the experimental results, with $R^{2}$ values closer to 90%. Specifically, for total air voids a linear plus interaction regression model was used with a $R^{2}$ value of 93.84%. A linear plus interaction regression model was used also for the interconnected air voids, with a $R^{2}$ value of 96.11%. Concerning the raveling resistance, the particle loss in dry and wet conditions was fitted using full quadratic regression models. In this case, $R^{2}$ values of 89.93% and 90.02%, respectively, were obtained. Similarly, a full quadratic regression equation was used to model the binder drainage, with the $R^{2}$ being equal to 89.53%. As for the mixtures using PMB 45/80-65, full quadratic regression models were applied to total air voids, interconnected air voids, particle loss in wet conditions and binder drainage, with *R*^2^ values of 64.86%, 30.02%, 80.04%, and 84.00%, respectively. In the case of particle loss in dry conditions, a linear regression model was applied with an $R^{2}$ value of 67.07%.

Figure 10, shows the graphs where $T_{AV},\;I_{AV},\;PL_{DRY},\;PL_{WET}$, and *BD* response variables were obtained experimentally and those predicted by the regression model for each binder type are compared. In the case of the mixtures with 50/70 penetration grade binder, predicted and experimental values are slightly closer to each other as compared to the case of the mixtures with PMB 45/80-65. As an example, the mean errors for the total air voids were of 1.61% and 2.01% when 50/70 penetration grade binder and PMB 45/80-65 were used, respectively. For the functionality responses, results suggest that the deviation between experimental data and regression models was minimal, with errors lower than 5%. However, the errors in the mechanical performance responses were in the range between 10% and 20%.

### 4.3. CRITIC Method

As said before, the CRITIC methodology is employed in this research for the purpose of finding out the weights of each criterion. The weights assigned to each response variable are based on the contrast intensity and conflict assessment of the decision making problem [55]. According to the methodology, the decision matrix is firstly normalized using Equation (13), as shown in Table 8. The standard deviation (*SD*) values for all the criteria are also calculated. The correlation coefficients of the different response variables were then calculated (Table 9). Finally, the weights of the different response variables were determined with the help of Equations (14)–(16), as shown in Table 10.

As can be seen in Table 10, total air voids and interconnected air voids have similar weights, which is due to the high correlation that exists between these two variables. On the other hand, particle loss in dry and wet conditions have the highest weights, with values of 0.24 and 0.25, respectively, suggesting that raveling resistance have a notable incidence in the overall performance of the PA mixture. Finally, the weight assigned to binder drainage was equal to 0.17, almost equal than $T_{AV}$ and $I_{AV}$ weights. As is well known, when weights are assigned equally, a subjective bias is involved in the decision-making process. To deal with this, CRITIC approach defines the criteria weightage in an objective manner, attempting to reveal the intensity of the contrast in the decision making problem [62]. 

### 4.4. TOPSIS Method

In this research, the Taguchi methodology was applied for the optimization of the single responses (e.g., total air voids, interconnected air voids, etc.) in the same way that other experimental design methods might have been used such as the central composite design, the response surface method or the full factorial design. Moreover, in this study more than one response was evaluated and hence, it is necessary to transform the multiple response variables into one single response variable. Therefore, TOPSIS methodology was employed as a multi-criteria decision-making technique built into the Taguchi experiment design method. 

Once the weights of the different response variables were calculated by applying the CRITIC approach, closeness comparative coefficient (*CCC*) for each design of experiments was determined employing Equations (4)–(11). Table 11 shows the weighted normalizes decision matrix for each response variable, with higher values of *CCC* indicating more optimum conditions. In this sense, the design ranked number 1 corresponds to the best combination of input parameters among all the set of experiments carried out. The positive ideal solution (PIS) values for each response is as follows: $V_{TAV}^{+}=0.0491$, $V_{IAV}^{+}$ = 0.0499, $V_{PL-dry}^{+}=0.0149$, $V_{PL-wet}^{+}=0.0123$ and $V_{BD}^{+}=0.0000$. Similarly, the negative ideal solution (NIS) values for each response is $V_{TAV}^{-\;}=0.0370$, $V_{IAV}^{-}=0.0324$, $V_{PL-dry}^{-}=0.1162$, $V_{PL-wet\;}^{-}=0.1410$ and $V_{BD}^{-}=0.1480$. After PIS and NIS were calculated, experiment designs were ranked based on *CCC* scores (Table 11). The experimental design number 17 resulted the best design, with response values of 20.22%, 14.15%, 4.77%, 5.26%, and 0.05% for $T_{AV},\;I_{AV},\;PL_{DRY},\;PL_{WET}\;$ and *BD*, respectively. This design involves the use of polymer modified binder with 0.15% fiber content and 5.0% binder content. On the other hand, experimental design number 3 was found to be the design with the lowest *CCC* value and hence, the last potential choice. Overall, the preference ranking of experimental designs can be given as 17 > 18 > 15 > 13 > 11 > 14 > 16 > 9 > 10 > 5 > 6 > 2 > 12 > 1 > 8 > 7 > 4 > 3.

*CCC* score values obtained via CRITIC-TOPSIS based Taguchi methodology were also used to calculate the main effects plots for SNR and main effect plots for means, as shown in Figure 11.

The type of binder seems to have the greatest impact on the SNR and means values. As can be observed in the ranking, the first seven experimental designs were about mixtures using PMB. On the other hand, good results were observed in terms of functionality and durability for the mixtures using 50/70 penetration grade binder. For example, mixtures corresponding to design number 5, with 0.05% fiber content and a 5.0% binder content, exhibited particle loss values in dry and wet conditions of 7.90% and 15.71%, respectively. According to the scientific literature, values lower than 20% and 35% are recommend in $PL_{DRY}$ and $PL_{WET}$ tests [2,36,37]. This mixture also shows a proper air void content of approximately 20% and does not present binder drainage problems, as it obtained a drain down value lower than 0.3%, the limit recommended in the literature [4].

When analyzing the *CCC* score values, trends indicate that low values of binder content and high values of fiber content clearly affect the overall performance of mixtures using 50/70 penetration grade bitumen, as can be observed in Figure 12. Likewise, all *CCC* values were below 0.80 in the case of mixtures using 50/70 conventional binder with the exception of design number 9, whose *CCC* value was 0.82. Moreover, design number 5 scored well after design number 9 with a value of 0.76. This experimental design exhibited lower values of particle loss in dry and wet conditions while maintaining admissible values of total and interconnected air voids. Besides in the case of the binder, drainage in this mixture was not observed. Therefore it could be considered as a proper mixture design. Finally, based on SNR, the TOPSIS approach suggests that the optimum conditions were identified for a binder type factor equal to PMB, fiber content factor of 0.05% and binder content factor of 5.0%.

As with the individual responses, a regression analysis was applied for the modeling of the *CCC* values and the analysis of the interaction effects between input parameters and the overall *CCC* response. A linear plus interaction predictive equation with a *p*-value of 0.004 significant effect was selected. The equation for *CCC* are given as follows:(31)CCC=1.128+0.2089∗BT−10.84 ∗FC (%)−0.1049∗BC(%)+2.268∗FC(%)∗BC(%). 

The graph given in Figure 13 shows the comparison between the *CCC* response obtained through the CRITIC-TOPSIS methodology and the *CCC* values from the regression model developed. The $R^{2}$ for the model obtained was 66.43% and the mean error between the *CCC* values calculated via CRITIC-TOPSIS and the model developed was of 11.78%. According to the analysis of variance (Table 12), the type of binder has a significant effect as well was the fiber-binder interaction. In other words, the overall performance of a PA mixture is linked to the proper quantities of fiber and binder, depending on the type of binder.

## 5. Conclusions

This study presented the CRITIC-TOPSIS based on the Taguchi methodology aimed at investigating the impact of different parameters on the mechanical and functional performance of fiber reinforced porous asphalt mixtures with aramid and polyolefin fibers. A series of experiments were carried out based on the L18 Taguchi orthogonal array, and the optimal responses were identified for the total air voids, interconnected air voids, particle loss in dry conditions, particle loss in wet conditions, and binder drainage. Signal to Noise Ratio values obtained from the Taguchi design made it possible to determine the optimal levels of the control factors for the different response variables. In addition, regression models were performed with the different responses in order to evaluate the binder-fiber interaction effects as a function of the type of bitumen. Since multiple responses were obtained, a multi objective optimization was performed through the CRITIC-TOPSIS methodology. Unlike other studies that assign equal weights to the different responses, the CRITIC approach was employed in this study to find the objective criteria weights. With TOPSIS, the criteria weights were taken into account to provide a preference ranking for all the designs of experiments. Based on the results obtained, the following conclusions can be drawn:In terms of functionality, the binder content is the most influential factor on the total and interconnected air voids of the mixture.Concerning the durability of the mixture, the optimum $PL_{dry}$ response based on Signal to Noise Ratio values is obtained when employing a polymer modified binder, a 0.05% fiber content, and a 5.5% binder content. The contribution of the fiber content is less significant when a polymer modified binder is used instead of a conventional binder.PA mixtures with a 50/70 penetration grade binder and 0.05% fiber content improve in a similar way to PA mixtures with a polymer modified binder. As for the raveling resistance, the addition of fibers reduces the particle loss in dry conditions regardless of the amount of bitumen employed. However, when it comes to the particle loss in wet conditions, a higher binder content seems to be necessary to properly coat the fibers and hence to guarantee a higher durability under the action of water.The use of fibers in the PA mixtures not only contributed to positively mitigating the binder drainage, but also to reinforcing the mixture without compromising its functionality.The best alternative according to the TOPSIS method is the design number 17. This design corresponds to the use of a polymer modified binder, 0.15% fiber content, and 5.0% binder content. Although the first few positions of the order of preference refers to experiments with mixes using polymer modified binder, good results can be also obtained using a conventional binder as long as the proper proportions of fibers are applied.The CRITIC-TOPSIS based Taguchi can be considered a useful tool for the evaluation of the impact of different admixtures on different responses, as well as for the optimization of multiple responses simultaneously. It is recommended to apply this novel methodology to other composites of materials.

## Figures and Tables

**Figure 1 materials-12-03789-f001:**
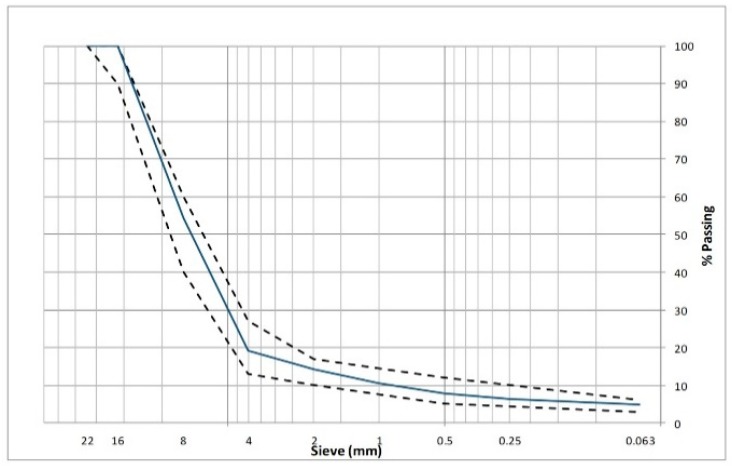
Gradation curve of the PA16 mixture.

**Figure 2 materials-12-03789-f002:**
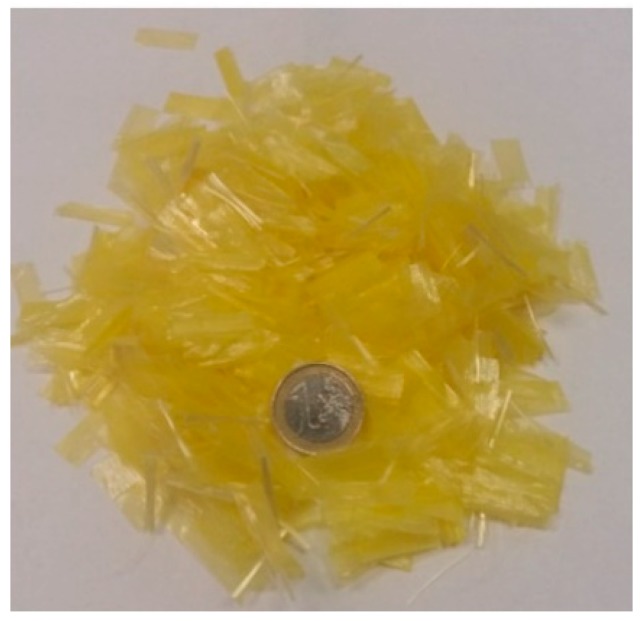
Blend of polyolefin and aramid (POA) fibers.

**Figure 3 materials-12-03789-f003:**
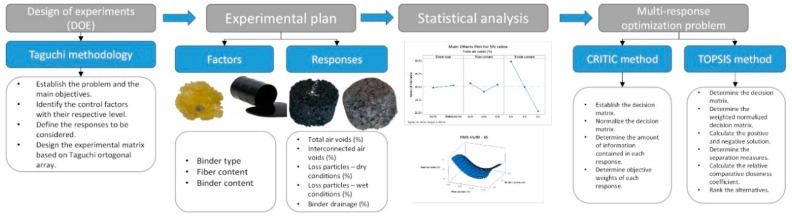
Structured framework proposed for the multi-objective optimization.

**Figure 4 materials-12-03789-f004:**
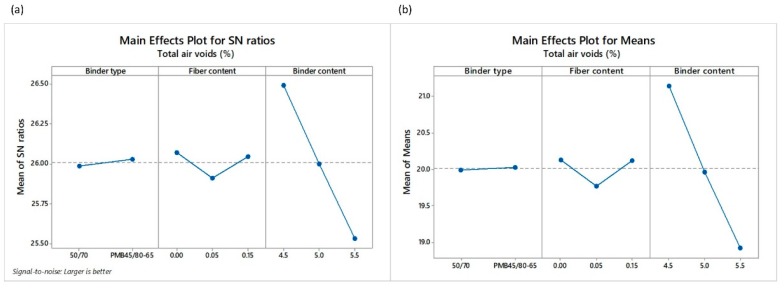
Main effects plots of (**a**) SNR and (**b**) means of the total air voids $T_{AV}$.

**Figure 5 materials-12-03789-f005:**
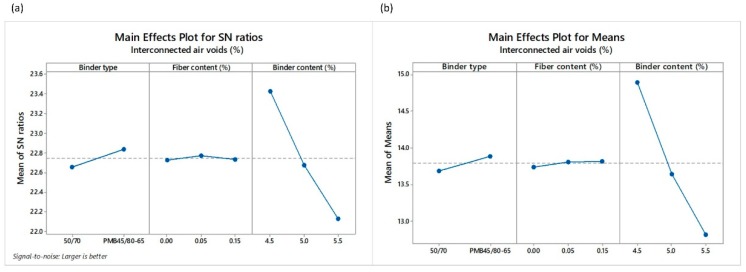
Main effects plots of (**a**) SNR and (**b**) means of the interconnected air voids *I_AV_*.

**Figure 6 materials-12-03789-f006:**
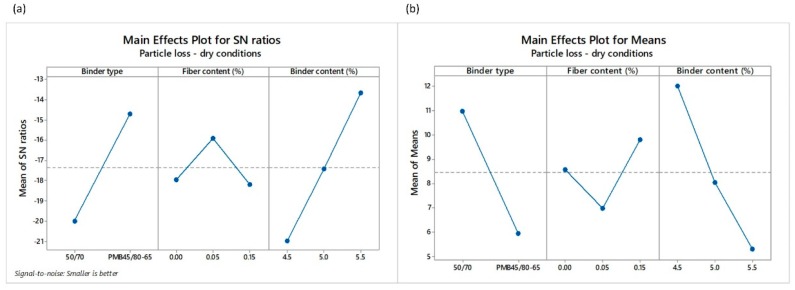
Main effects plots of (**a**) SNR and (**b**) means of the particle loss in dry conditions $PL_{DRY}$.

**Figure 7 materials-12-03789-f007:**
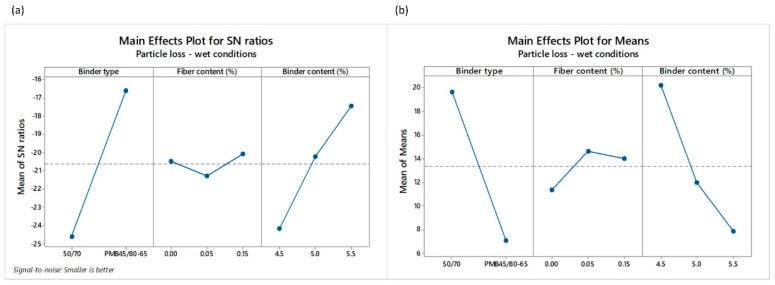
Main effects plots of (**a**) SNR and (**b**) means of the particle loss in wet conditions $PL_{WET}.$

**Figure 8 materials-12-03789-f008:**
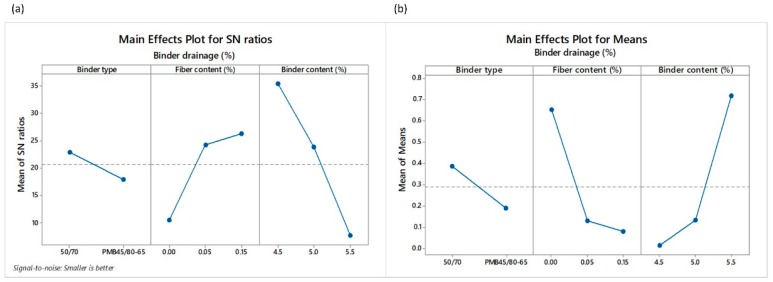
Main effects plots of (**a**) SNR and (**b**) means of the binder drainage (*BD*).

**Figure 9 materials-12-03789-f009:**
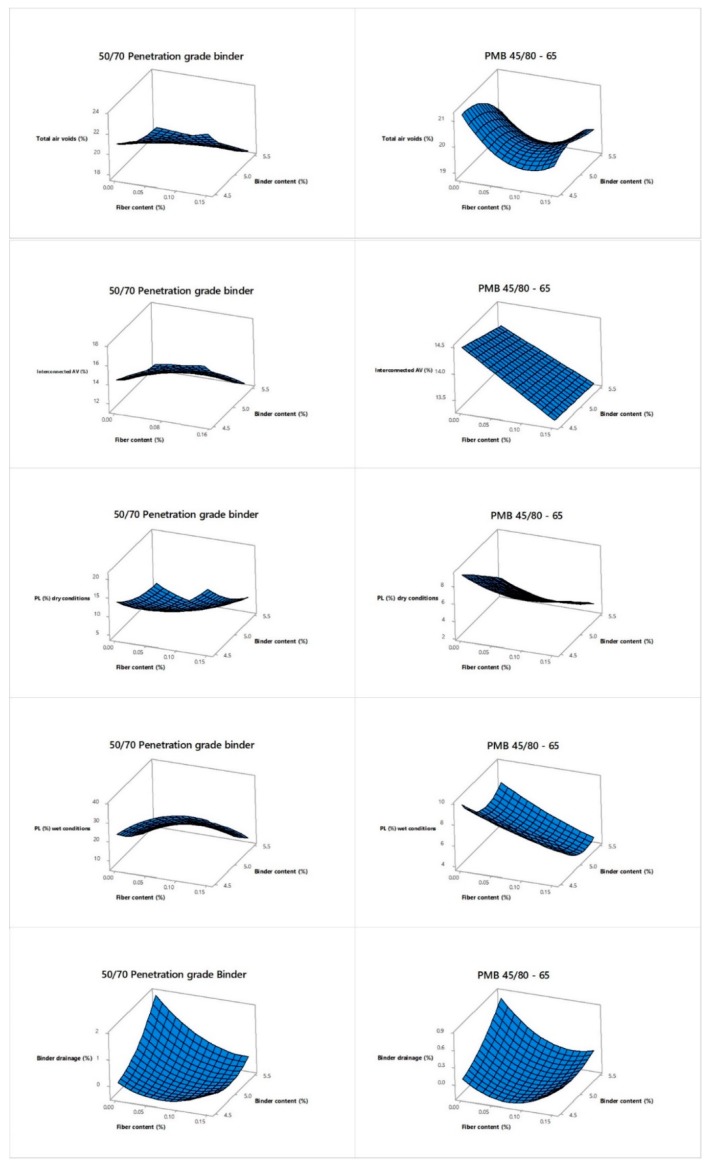
Interaction effect of fiber content and binder content as a function of the binder type.

**Figure 10 materials-12-03789-f010:**
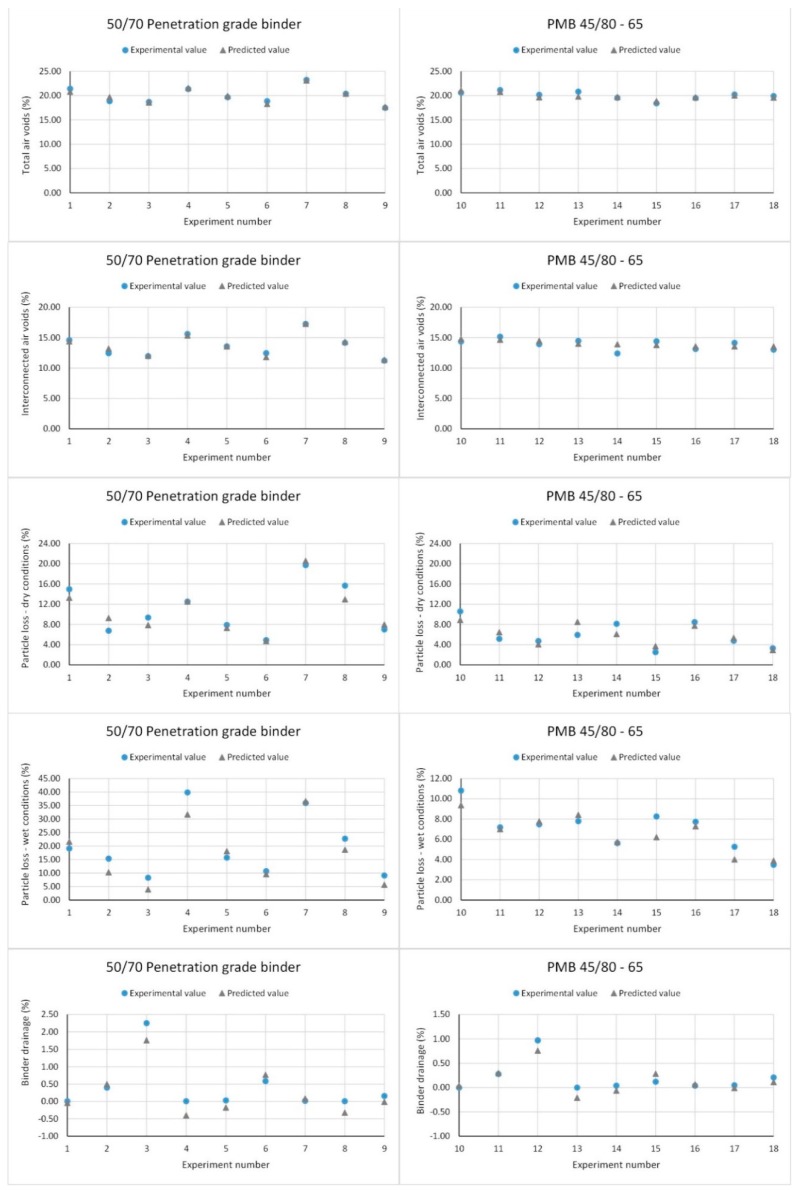
Experimental vs. predicted values of the different response variables.

**Figure 11 materials-12-03789-f011:**
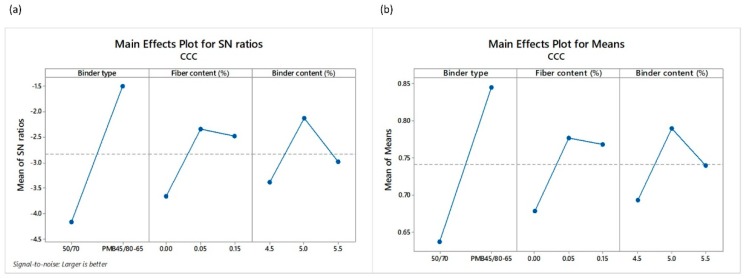
Main effects plots of (**a**) SNR and (**b**) means of the *CCC* values.

**Figure 12 materials-12-03789-f012:**
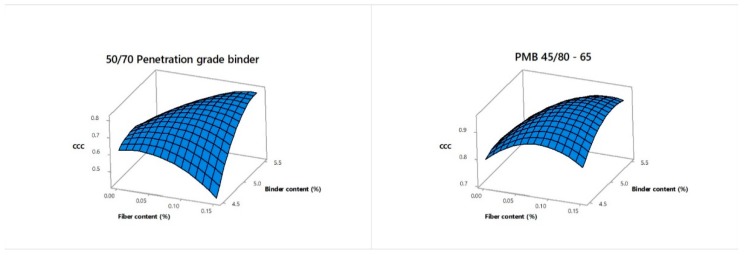
Interaction effect of fiber content and binder content as a function of the *CCC* score value for a 50/70 penetration grade binder (left) and a PMB 45/80-65 (right).

**Figure 13 materials-12-03789-f013:**
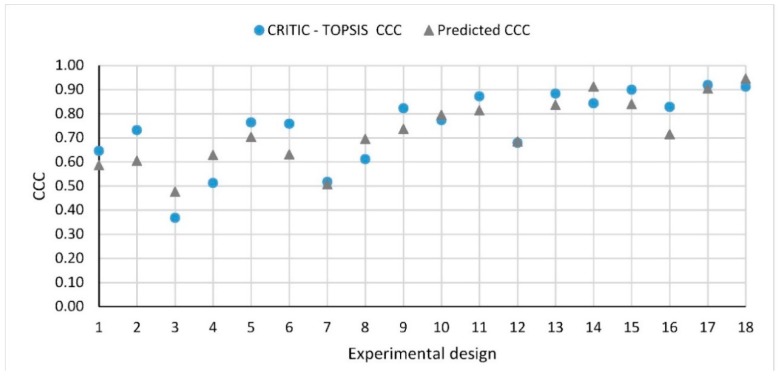
Comparison between the calculated *CCC* response and the predicted model.

**Table 1 materials-12-03789-t001:** Input parameters and their corresponding levels.

Input Parameter	Notation	Level 1	Level 2	Level 3
Binder type	BT	50/70	PMB	-
Fiber content (%)	FC	0.00	0.05	0.15
Binder content (%)	BC	4.5	5.0	5.5

**Table 2 materials-12-03789-t002:** Full factorial design with Taguchi orthogonal array L18.

Design	Binder Type	Fiber Content	Binder Content
1	50/70	0.00	4.50
2	50/70	0.00	5.00
3	50/70	0.00	5.50
4	50/70	0.05	4.50
5	50/70	0.05	5.00
6	50/70	0.05	5.50
7	50/70	0.15	4.50
8	50/70	0.15	5.00
9	50/70	0.15	5.50
10	PMB45/80-65	0.00	4.50
11	PMB45/80-65	0.00	5.00
12	PMB45/80-65	0.00	5.50
13	PMB45/80-65	0.05	4.50
14	PMB45/80-65	0.05	5.00
15	PMB45/80-65	0.05	5.50
16	PMB45/80-65	0.15	4.50
17	PMB45/80-65	0.15	5.00
18	PMB45/80-65	0.15	5.50

**Table 3 materials-12-03789-t003:** Physical properties of coarse (ophite) and fine (limestone) aggregates.

Characteristic	Value	Standard	Specification
Coarse Aggregate			
Specific Weight (g/cm^3^)	2.794	EN 1097-6	-
Water absorption (%)	0.60	EN 1097-6	<1%
L.A abrasion (%)	15	EN 1097-2	≤15%
Slab Index (%)	<1%	EN 933-3	≤20%
Polishing Value	60	EN 1097-8	≥56
Fine Aggregate			
Specific Weight (g/cm^3^)	2.724	EN 1097-6	-
Sand Equivalent	78	EN 933-8	>55

**Table 4 materials-12-03789-t004:** Main properties of the binders used.

Binder	Test	Standard Method	Value
50/70	Penetration at 25 °C (mm/10)	EN 1426	57.00
	Specific Gravity	EN 15326	1.04
	Softening point (°C)	EN 1427	51.60
	Fraass brittle point (°C)	EN 12593	−13.00
PMB 45/80-65	Penetration at 25 °C (mm/10)	EN 1426	49.50
	Specific Gravity	EN 15326	1.03
	Softening point (°C)	EN 1427	72.30
	Fraass fragility point (°C)	EN 12593	−15.00
	Ductility force at 5 °C (J/cm^2^)	EN 13589	3.11
	Elastic recovery at 25 °C (%)	EN 13398	90.00

**Table 5 materials-12-03789-t005:** Characteristics of POA fibers.

Fiber	Aramid	Polyolefin
Form	Monofilament	Serrated
Color	Yellow	Yellow
Density (g/cm^3^)	1.44	0.91
Length (mm)	19	19
Tensile Strength (MPa)	2758	483
Decomposition temperature (°C)	> 450	157
Acid/Alkali Resistance	Inert	Inert

**Table 6 materials-12-03789-t006:** L18 Taguchi orthogonal array response variables.

Design	Total Air Voids (*T_AV_*)		Interconnected Air Voids (*I_AV_*)		Particle Loss-Dry Condition (*PL_DRY_*)		Particle Loss-Wet Condition (*PL_WET_*)		Binder Drainage (*BD*)
	mean	*SD*	mean	*SD*	mean	*SD*	mean	*SD*	
1	21.39	0.75	14.59	1.32	14.96	1.99	19.12	5.51	0.01
2	18.85	0.14	12.45	0.70	6.76	2.65	15.32	3.20	0.40
3	18.68	1.12	11.94	1.48	9.37	0.75	8.28	2.03	2.25
4	21.36	0.35	15.59	1.01	12.52	1.99	39.85	10.23	0.01
5	19.67	0.40	13.57	0.61	7.90	4.27	15.71	1.89	0.03
6	18.85	0.14	12.45	0.70	4.90	1.68	10.70	1.40	0.59
7	23.22	0.22	17.26	0.38	19.71	2.01	35.95	5.05	0.02
8	20.38	0.88	14.14	1.06	15.66	1.86	22.74	3.15	0.01
9	17.49	1.30	11.22	0.99	7.01	1.39	9.08	2.15	0.16
10	20.59	1.89	14.36	2.22	10.57	4.80	10.81	3.54	0.00
11	21.12	0.40	15.16	0.80	5.16	2.77	7.19	1.68	0.28
12	20.18	2.18	13.93	2.91	4.73	0.78	7.49	1.72	0.97
13	20.81	2.14	14.47	2.66	5.94	2.20	7.80	3.52	0.00
14	19.54	1.88	12.39	2.80	8.12	5.19	5.62	0.26	0.04
15	18.42	2.51	14.39	3.43	2.52	0.96	8.25	2.47	0.12
16	19.50	1.14	13.12	0.91	8.47	3.70	7.73	0.45	0.04
17	20.22	0.17	14.15	0.11	4.77	1.02	5.26	0.76	0.05
18	19.91	1.03	13.03	0.97	3.30	0.34	3.48	0.62	0.21

**Table 7 materials-12-03789-t007:** Analysis of variance for *T_AV_*, *I_AV_*, *PL_DRY_*, *PL_WET_* and *BD*.

Variance Source	Degree of Freedom (DoF)	Adj SS	Adj MS	*F*-Value	Contribution (%)
Total air voids (%)					
Fiber content (%)	2	0.783	0.392	0.42	3.12
Binder content (%)	2	20.565	10.282	10.98	81.95
Error	4	3.747	0.937		14.93
Total	8	25.095			100.00
Interconnected air voids (%)					
Fiber content (%)	2	2.339	1.170	1.32	7.88
Binder content (%)	2	23.794	11.897	13.48	80.21
Error	4	3.531	0.883		11.90
Total	8	29.664			100.00
Particle loss—dry conditions					
Fiber content (%)	2	50.220	25.112	3.01	25.22
Binder content (%)	2	115.440	57.722	6.91	57.98
Error	4	33.420	8.354		16.79
Total	8	199.090			100.00
Particle loss—wet conditions					
Fiber content (%)	2	131.500	65.760	1.76	12.66
Binder content (%)	2	757.900	378.970	10.16	72.97
Error	4	149.100	37.290		14.36
Total	8	1038.600			100.00
Binder drainage					
Fiber content (%)	2	1.157	0.579	1.68	27.21
Binder content (%)	2	1.719	0.860	2.5	40.43
Error	4	1.375	0.344		32.34
Total	8	4.252			100.00

**Table 8 materials-12-03789-t008:** Normalized decision matrix for the CRITIC method.

Design	*T_AV_* (%)	*I_AV_* (%)	*PL_DRY_* (%)	*PL_WET_* (%)	*BD* (%)
1	0.68	0.56	0.28	0.57	1.00
2	0.24	0.20	0.75	0.67	0.82
3	0.21	0.12	0.60	0.87	0.00
4	0.68	0.72	0.42	0.00	1.00
5	0.38	0.39	0.69	0.66	0.99
6	0.24	0.20	0.86	0.80	0.74
7	1.00	1.00	0.00	0.11	0.99
8	0.50	0.48	0.24	0.47	1.00
9	0.00	0.00	0.74	0.85	0.93
10	0.54	0.52	0.53	0.80	1.00
11	0.63	0.65	0.85	0.90	0.88
12	0.47	0.45	0.87	0.89	0.57
13	0.58	0.54	0.80	0.88	1.00
14	0.36	0.19	0.67	0.94	0.98
15	0.16	0.53	1.00	0.87	0.95
16	0.35	0.32	0.65	0.88	0.98
17	0.48	0.49	0.87	0.95	0.98
18	0.42	0.30	0.95	1.00	0.91
*SD*	0.23	0.24	0.27	0.28	0.25

**Table 9 materials-12-03789-t009:** Correlation coefficients of the different response variables.

	*T_AV_* (%)	*I_AV_* (%)	*PL_DRY_* (%)	*PL_WET_* (%)	*BD* (%)
*T_AV_* (%)	1.00	0.89	−0.63	−0.59	0.34
*I_AV_* (%)	0.89	1.00	−0.50	−0.62	0.40
*PL_DRY_* (%)	−0.63	−0.62	1.00	0.79	−0.16
*PL_WET_* (%)	−0.59	−0.62	0.79	1.00	−0.24
*BD* (%)	0.34	0.40	−0.16	−0.24	1.00

**Table 10 materials-12-03789-t010:** Weights of the different response variables.

Criteria	*C_j_*	*W_j_*
*T_AV_* (%)	0.93	0.18
*I_AV_* (%)	0.92	0.17
*PL_DRY_* (%)	1.25	0.24
*PL_WET_* (%)	1.31	0.25
*BD* (%)	0.90	0.17

**Table 11 materials-12-03789-t011:** Weighted normalized response, *CCC* values and final ranking.

Design No.	Weighted Normalized Values									
	*T_AV_* (%)	*I_AV_* (%)	*PL_DRY_* (%)	*PL_WET_* (%)	*BD* (%)	S_i_^+^	S_i_^−^	S_i_^+^ + S_i_^−^	*CCC*	Rank
1	0.045	0.042	0.088	0.068	0.001	0.09	0.17	0.26	0.65	14
2	0.040	0.036	0.040	0.054	0.026	0.06	0.16	0.22	0.73	12
3	0.040	0.035	0.055	0.029	0.148	0.16	0.09	0.25	0.37	18
4	0.045	0.045	0.074	0.141	0.001	0.14	0.15	0.29	0.51	17
5	0.042	0.039	0.047	0.056	0.002	0.06	0.18	0.23	0.76	10
6	0.040	0.036	0.029	0.038	0.039	0.05	0.16	0.21	0.76	11
7	0.049	0.050	0.116	0.127	0.001	0.15	0.16	0.32	0.52	16
8	0.043	0.041	0.092	0.080	0.001	0.10	0.16	0.27	0.61	15
9	0.037	0.032	0.041	0.032	0.011	0.04	0.19	0.23	0.82	8
10	0.044	0.042	0.062	0.038	0.000	0.06	0.19	0.24	0.77	9
11	0.045	0.044	0.030	0.025	0.018	0.03	0.19	0.22	0.87	5
12	0.043	0.040	0.028	0.026	0.064	0.07	0.14	0.21	0.68	13
13	0.044	0.042	0.035	0.028	0.000	0.03	0.21	0.23	0.88	4
14	0.041	0.036	0.048	0.020	0.003	0.04	0.20	0.24	0.84	6
15	0.039	0.042	0.015	0.029	0.008	0.02	0.20	0.23	0.90	3
16	0.041	0.038	0.050	0.027	0.003	0.04	0.20	0.24	0.83	7
17	0.043	0.041	0.028	0.019	0.003	0.02	0.21	0.23	0.92	1
18	0.042	0.038	0.019	0.012	0.014	0.02	0.21	0.23	0.91	2

**Table 12 materials-12-03789-t012:** Analysis of variance of the regression model developed for *CCC*.

Source	DF	Adj SS	Adj MS	*F*-Value	*P*-Value	Significance
Model	4	0.280	0.070	6.430	0.004	Significant
Linear	3	0.226	0.075	6.930	0.005	Significant
Binder type	1	0.196	0.196	18.020	0.001	Significant
Fiber content (%)	1	0.018	0.018	1.610	0.227	
Binder content (%)	1	0.013	0.013	1.150	0.303	
2-Way Interaction	1	0.060	0.060	5.510	0.035	Significant
Fiber content (%) *Binder content (%)	1	0.060	0.060	5.510	0.035	Significant
Error	13	0.142	0.011			
Total	17	0.422

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
