# Peer review of "Multi-Response Optimization of Porous Asphalt Mixtures Reinforced with Aramid and Polyolefin Fibers Employing the CRITIC-TOPSIS Based on Taguchi Methodology"

_materials, 2019, doi:10.3390/ma12223789_

Round 1

Reviewer 1 Report

Comments and Suggestions for Authors

In this work, the authors investigated statistically the influence of three control factors such as binder type, fiber content and binder content in terms of five responses such as total air voids, interconnected air voids, particle loss in dry conditions, particle loss in wet conditions and binder drainage. The experiments were conducted based on Taguchi L18 orthogonal array. Multiple regression models were employed to predict the responses of the experiments. A multi-objective optimization was performed by employing Criteria Importance through Criteria Inter-Correlation (CRITIC) and Technique for Order Preference by Similarity to Ideal Solution (TOPSIS) methodologies. The article may be with reliable data. I suggest the article be accepted after minor revision, some suggestions are as follows:

The experiments were conducted based on Taguchi L18 orthogonal array (max. 8 control factors), but only 3 control factors were investigated. Why did the authors not use Taguchi L9 orthogonal array (less experiment) or L16 orthogonal array (more levels)? The "Abstract" mentioned the promising results were obtained. But they were not discussed in the content. How were they obtained? In section 4.1, for the correlation coefficients, e.g. 89% (pp. 10) and 79% (pp. 11), how were the values determined? In Table 7, why is the input parameter of “binder type” not in the analysis of variance? For the regression analyses, as for the mixtures using PMB 45/80 – 65, full quadratic regression models was applied to interconnected air voids with R2 values of only 30.02%, please check it. For analyzing the CCC score values, in the case of mixtures using 50/70 conventional binder, the design 5 scored with a value of 0.82, but it is 0.76 in Table 11, why? Are they promising results? The authors are suggested to refine the conclusion part. Only main findings are expected. Some outlooks could be added to offer guidance for further study.

Author Response

The experiments were conducted based on Taguchi L18 orthogonal array (max. 8 control factors), but only 3 control factors were investigated. Why did the authors not use Taguchi L9 orthogonal array (less experiment) or L16 orthogonal array (more levels)?

Since three control factors were investigated, a Taguchi L9 orthogonal array or L16 orthogonal array might have been used, as suggested by the reviewer. However, since in this case one of the factor is categorical (binder type) it was not possible to reduce the number of experiments and hence, the full factorial L18 orthogonal array had to be used.

The "Abstract" mentioned the promising results were obtained. But they were not discussed in the content. How were they obtained?

As suggested by the reviewer, some extra lines (472 – 475) were added to the manuscript in order to provide further information about the results with the experimental design number 5. Likewise, the term “promising results” was changed to “good results”.

In section 4.1, for the correlation coefficients, e.g. 89% (pp. 10) and 79% (pp. 11), how were the values determined?

The Pearson coefficient was determined between responses. As it was not mentioned in the manuscript, the word “Pearson” has been added in those cases where the term “correlation coefficient” appears in the text (lines 301 and 324). It is well known that the Pearson correlation coefficient is a measure of the linear correlation between two different variables and hence, we honestly did not believe that giving further explanation is necessary.

In Table 7, why is the input parameter of “binder type” not in the analysis of variance? For the regression analyses, as for the mixtures using PMB 45/80 – 65, full quadratic regression models was applied to interconnected air voids with R2 values of only 30.02%, please check it.

The analysis of variance was done only with 50/70 penetration graded binder since good match were found between fiber content and binder content with this type of binder. In addition, the best regression models and predictive equations were found for this type of binder instead of a polymer modified binder. As suggested by the reviewer, the R2 values of the full quadratic regression models applied to the interconnected air voids response has been carefully checked. Even with the full quadratic regression model, 30.02% was the maximum value obtained.

For analyzing the CCC score values, in the case of mixtures using 50/70 conventional binder, the design 5 scored with a value of 0.82, but it is 0.76 in Table 11, why? Are they promising results?

Thank you so much for the observation. It was indeed a mistake. The real score corresponds to 0.76, as shown in table 11. Accordingly, the paragraph was rewritten (lines 469 – 472). This design is considered with good results from functional and mechanical point of view since low values of particle loss was obtained while maintaining a proper functionality in terms of total and interconnected air voids.

The authors are suggested to refine the conclusion part. Only main findings are expected. Some outlooks could be added to offer guidance for further study.

As suggested by the reviewer, only the most relevant conclusions have been kept. Since this research has two important components, the experimental aspect but also the statistical and decision-making methodology, one conclusion has been specifically devoted to recommend the use of this novel multi-criteria decision making methodology in other composites of materials.

Reviewer 2 Report

The paper presents a research on the impact of different parameters on the mechanical and functional performance of fiber reinforced porous asphalt mixtures with aramid and polyolefin fibers. Taguchi L18 orthogonal array was used in the design of experiments and the input factors selected were the binder type, the fiber content and the binder content. The paper presents multiple regression models regarding the prediction of the responses of the experiments. CRITIC-TOPSIS based Taguchi methodology was employed in this research as a multi-objective optimization tool in order to find the most optimized single responses regarding the functionality and durability requirements of the material.

The paper focus a very interesting topic on PA mixtures reinforced with fibers and the research was well conducted. The paper is well written and the structure is adequate. Conclusions are clear.

Some minor corrections are recommended:

Despite this abbreviation being established in European standard EN 13108-7, it is recommended to use in the title the complete designation: Porous Asphalt It is recommended to clarify the objective of the paper. The present abstract is more related to methodology and conclusions. What is the reason to use bolds when refereeing tables, figures and equations along the text? It is recommended to use PA abbreviation in the first paragraph of section 3.1 Use spaces when referring EN 933 – 8 in Table 3. Please, check other cases. In section 3.3, when referring the binder drainage test, please clearly specify the method In Figure 3, the images associates to statistical analysis are impossible to read. In Figure 5, the symbol for interconnected air voids should be formated In Figure 9, the images are very small and it is difficult to read the legends in some cases In section 5, please use bullet in first conclusion In acknowledgments section, the first sentence should be included in a funding section (not an acknowledgment). Other sections should be included (author contributions, conflicts of interest...) Check in general the references format (order of name and initials of authors, titles format). References 35 and 62 are duplicated.

Author Response

Despite this abbreviation being established in European standard EN 13108-7, it is recommended to use in the title the complete designation: Porous Asphalt It is recommended to clarify the objective of the paper. The present abstract is more related to methodology and conclusions.

As proposed by the reviewer, the complete designation of the PA mixture (porous asphalt) has been used in the title. In the abstract, we focused on the novel methodology and the main conclusions with the sole intention of making it shorter and more straightforward and readable.

What is the reason to use bolds when refereeing tables, figures and equations along the text?

We used bolds only to distinguish the references to tables, figures and equations from the normal text, thus making it more readable. Of course, the manuscript can adapted to the format of the journal.

It is recommended to use PA abbreviation in the first paragraph of section 3.1

Thank you for the recommendation. It was modified accordingly.

Use spaces when referring EN 933 – 8 in Table 3. Please, check other cases. In section 3.3, when referring the binder drainage test, please clearly specify the method In Figure 3.

Thank you for your suggestion. We checked the tables carefully. Concerning the binder drainage test, we included in the text that the mesh basket drain down test was used according to the European standard EN 12697 – 18. We did not include it in figure 3 since in the illustration we are referring to the responses obtained instead of the tests carried out.

The images associates to statistical analysis are impossible to read. In Figure 5, the symbol for interconnected air voids should be formated In Figure 9, the images are very small and it is difficult to read 

The symbol for interconnected air voids (IAV) was properly written in the original manuscript sent to the journal (as in figure 4). We hope that the journal team writes it well in the final version. Otherwise, we will let they know about it. As for the quality of the images, a definition of 300 ppp has been set for all the figures in order to make them more readable and comprehensible.

In section 5, please use bullet in first conclusion.

Corrected.

 In acknowledgments section, the first sentence should be included in a funding section (not an acknowledgment). Other sections should be included (author contributions, conflicts of interest...) Check in general the references format (order of name and initials of authors, titles format). References 35 and 62 are duplicated.

As suggested by the reviewer, a funding section has been added and the acknowledgments section has been revised accordingly as well as the author’s contributions. The references were checked and duplicated references were removed.